# Comparative Analysis of Transcriptional Regulation Patterns: Understanding the Gene Expression Profile in *Nucleocytoviricota*

**DOI:** 10.3390/pathogens10080935

**Published:** 2021-07-24

**Authors:** Fernanda Gil de Souza, Jônatas Santos Abrahão, Rodrigo Araújo Lima Rodrigues

**Affiliations:** Laboratório de Vírus, Departamento de Microbiologia, Instituto de Ciências Biológicas, Universidade Federal de Minas Gerais, Belo Horizonte, Minas Gerais 31270-901, Brazil; nandags10@hotmail.com

**Keywords:** NCLDV, transcription, giant viruses, gene expression, evolution

## Abstract

The nucleocytoplasmic large DNA viruses (NCLDV) possess unique characteristics that have drawn the attention of the scientific community, and they are now classified in the phylum *Nucleocytoviricota*. They are characterized by sharing many genes and have their own transcriptional apparatus, which provides certain independence from their host’s machinery. Thus, the presence of a robust transcriptional apparatus has raised much discussion about the evolutionary aspects of these viruses and their genomes. Understanding the transcriptional process in NCLDV would provide information regarding their evolutionary history and a better comprehension of the biology of these viruses and their interaction with hosts. In this work, we reviewed NCLDV transcription and performed a comparative functional analysis of the groups of genes expressed at different times of infection of representatives of six different viral families of giant viruses. With this analysis, it was possible to observe a temporal profile of their gene expression and set of genes activated in specific phases throughout the multiplication cycle as a common characteristic of this group. Due to the lack of information regarding the transcriptional regulation process of this group of pathogens, we sought to provide information that contributes to and opens up the field for transcriptional studies of other viruses belonging to *Nucleocytoviricota*.

## 1. Introduction

The nucleocytoplasmic large DNA viruses (NCLDV) are characterized by sharing a set of conserved genes related to replication, transcription, and morphogenesis, a phenomenon that suggests these viruses have a common evolutionary origin. Subsequent analyses of sequenced genomes of isolated viruses belonging to new families have supported the monophyly of the NCLDV group [1]. Initial comparative genomic analyses have derived a set of 40 core genes common to NCLDV. Among these core genes, there are only a few conserved in all known viruses that belong to this group, including the DNA polymerase B family, helicase-primase, and the late transcription factor 3 [2,3,4]. This set of genes called “hallmark genes”, reconstructed as present in the common ancestor, seem to have origins from different sources, the majority being homologous to eukaryotic genes and a small part derived from bacteriophage genes [3].

Distinct viral families are part of the NCLDV group, including *Poxviridae*, *Asfarviridae*, *Iridoviridae*, *Phycodnaviridae*, *Ascoviridae*, *Mimiviridae*, and *Marseilleviridae*. NCLDV have recently been officially classified into a new phylum, namely *Nucleocytoviricota* [5]. Other recently discovered giant viruses such as pandoravirus, faustovirus, kaumoebavirus, cedratvirus, pithovirus, mollivirus, pacmanvirus, orpheovirus, and medusavirus have also been included as members of the NCLDV group, despite not being officially classified into any existent taxa [6,7,8,9,10,11,12,13,14]. This group has unique characteristics such as large particles and genomes, which encode proteins that have never been described in other viruses [15]. Another characteristic shared by NCLDV is the fact that they replicate entirely or partially in the host cell’s cytoplasm, in which some viral groups exhibiting little dependence on the host cell’s transcriptional machinery, such as poxviruses and mimiviruses [15,16]. The presence of a robust transcriptional apparatus in some NCLDV has raised discussion about the origin and evolution of these viruses and their genomes [16]. In addition, their gene transcription is temporally regulated, allowing sets of genes to be classified as early, intermediate, or late in accordance with the stage of infection when they are transcribed. A previous study has divided the genes of NCLDV into clusters of orthologous groups (NCVOG), many of which could be assigned to putative functional classes [1].

In this work, we reviewed NCLDV transcription and performed a comparative functional analysis according to what had been described previously for NCVOG as early, intermediate, and late genes expressed throughout the replication cycle of different viruses, considering representatives of six viral groups within *Nucleocytoviricota*. This analysis provides information on a transcriptional pattern of these viruses, with additional evidence for a common origin of this group of pathogens. Furthermore, understanding the regulation of gene expression throughout the replication cycle also improves our knowledge about the biology of giant viruses.

## 2. Temporal Regulation of Gene Expression

Transcriptional regulation involves a sequence of steps and although most of them have been studied extensively using static biochemistry, much about the real-time kinetics of transcription has not been completely elucidated [17]. As previously described, NCLDV have many characteristics in common. Among these characteristics, the transcription of their genes has a temporal profile, being classified as early, intermediate, and late; the genes expressed at each time point have different functions. The expression of each class of gene occurs in a cascade. This is due to the fact that the required transcription factors for the expression of each class of genes are the product of the genes previously expressed [18]. In that way, the products of some genes expressed early during the replication cycle of the virus will be required as transcription factors to induce the expression of other genes that will be expressed during the intermediate and later course of infection. Despite having a similar transcriptional regulation profile, the expression of the genes of each temporal class occurs at different times among NCLDV, due to the fact that their replication cycles last for different lengths of time. Nevertheless, the expression of genes from different time classes occurs at similar phases throughout the NCLDV multiplication cycle, that is, early, intermediate, and late during the course of the infection [16,18].

Such temporal classification has been possible due to the development of new techniques to quantify the expression levels of a large number of genes [19]. DNA microarrays were one of the first tools that allowed the large-scale study of the transcriptome. The technique is based on the hybridization of target strands on the complementary probe strand, allowing the identification of genes that are expressed at different times during the viral infection. A major limitation of this technique is to evaluate genes with low expression. Nevertheless, the advance of large-scale sequencing tools has allowed a more robust evaluation of the transcriptome of viruses. RNA sequencing (RNA-seq) has been widely used to study the gene expression profile of organisms and viruses, allowing a more in-depth comprehension of their transcriptional pattern. This technique provides complete sequencing of all expressed genes during the replication cycle of a virus, even if the genes have a low level of expression (depending on the coverage of the sequencing), allowing a better characterization of the virus’s transcriptome. A full description of the fundaments and possible usages of these techniques are beyond the scope of this review, but this topic has been extensively addressed elsewhere [20,21]. Although these tools are based on different technologies, both techniques make it possible to quantify the levels of gene expression and have had similar coverage in studies that have compared both techniques. Hence, they can be used in a comparison or as complementary methods in studies involving gene expression [19]. These techniques have been used to study the expression profile of different large and giant viruses, as will become apparent in the subsequent sections, providing important information about the biology of these viruses.

## 3. Transcription in *Nucleocytoviricota*

### 3.1. Poxvirus Gene Transcription

Among the viral families that comprise NCLDV, *Poxviridae* is by far the most studied group. Viruses belonging to this family have enveloped, ovoid-shaped particles that are 200 nm in diameter and 300 nm in length. Its genome consists of linear double-stranded DNA (dsDNA) of approximately 200 kilobase pairs (kbp), encoding 200 open reading frames (ORFs) (Table 1) [22,23]. This viral family comprises many human and other animal pathogens, including the one responsible for the most devastating disease that has affected humankind, the variola virus.

Viruses possess a great diversity in genome composition, structure, replication, and transcriptional strategies, which is manifested both in their biology and the host–virus interaction. Some DNA viruses with smaller genomes depend on host cell enzymes for transcription, including RNA polymerase [16]. However, some DNA viruses with larger genomes can encode their transcriptional apparatus, making them relatively independent of the host’s transcriptional machinery [24]. The complexity of the poxvirus genome instigated speculation that these viruses synthesize their genome independently of the host nucleus. The transcription of poxvirus genes follows a temporal profile and is regulated by promoter regions and transcriptional factors. Thus, the genes are classified into early, intermediate, and late, and they are activated in a cascade sequence [16,18,25].

Early transcribed genes encode proteins necessary for DNA replication, transcription of intermediate genes, and evasion of host defenses, while intermediate and late genes encode structural proteins that participate in the morphogenesis of viral progeny [18]. In addition, genes transcribed later encode components of the early transcriptional apparatus, present in the mature particle. After entry and release of the genetic content in the host cell cytoplasm, early transcription factors bind to AT-rich promoter sequences in upstream and downstream regions of the transcription start site, leaving the early genes available for viral RNA polymerase [18,43]. Subsequently, RNA polymerase is recruited, forming a complex with early transcription factors (VETF), termination factors, and poly(A) polymerases, which bind to RNA polymerase through the RAP94 domain [44]. The RAP94 domain, which is specific to the RNA polymerase packaged in the virion, is the key link between the RNA polymerase and VETF, being essential for gene expression [45]. It is interesting to note that there are no homologs of RAP94 outside poxviruses, suggesting the poxvirus early gene transcription system as an evolutionary outlier [18].

Transcription of intermediate and late genes occurs immediately after DNA synthesis is initiated. Vaccinia virus intermediate and late transcription require RNA polymerase to be synthesized after infection. Differently for early genes, the presence of RAP94 is not essential for these classes of genes [46]. Some viral proteins such as VITF-1, a subunit of RNA polymerase, as well as a viral capping enzyme and a heterodimer of VITF-3, have been identified as necessary for the transcription of intermediate genes [18,44,45]. Besides the viral proteins, a cellular component called VITF-2 was found to be necessary to complement in vitro viral transcription [46,47]. Intermediate promoter motifs regulate the expression of late transcription factor genes, consistent with the cascade model of regulation. For late gene transcription, the viral proteins encoded by *A1L*, *A2L G8R*, and *H5R* genes have been identified as essential. These proteins have been described as showing activity as late transcription factors [48]. As observed for poxviruses’ early transcription factors, none of the intermediate or late transcription factors have significant homologs in cellular proteins, highlighting the uniqueness of the viral regulatory system. As observed for intermediate genes, cellular components are suggested to participate in the regulation of late genes. Finally, in vitro assays suggest a role for a host factor originally called VLTF-X, and also the association of the TATA-box binding protein (TBP) for viral intermediate and late transcription [26,49]. The virus-encoded transcription factors may link the TBP to the viral RNA polymerase and place the initiator in the correct position near the active site for their respective promoters, thus regulating the gene expression [18].

Analyses performed with cells infected with cowpox virus (CPXV), monkeypox virus (MPXV), and vaccinia virus (VACV) have demonstrated that approximately 96% of the cellular transcripts did not change the expression profile. Of the cellular genes that changed the expression profile, 64.3%, 68.2%, and 70% had an increase in the expression profile modulated by infection by CPXV, MPXV, and VACV, respectively. Besides, a typical pattern of transcriptional modulation among the three viruses has been identified. Thus, of the 321 host cell transcripts modulated by MPXV infection, 241 (75.1%) are also modulated by CPXV and 148 (46.1%) by VACV [50]. This temporal profile of gene transcription seems to be a common characteristic not only among members of the family, but also of other viruses related phylogenetically to poxviruses, such as asfarviruses that infect mammals.

### 3.2. Asfarvirus Gene Transcription

The African swine fever virus (ASFV) is the only characterized member of the *Asfarviridae* family. This pathogen can infect pigs and wild boars. Its genome consists of linear dsDNA, with approximately 170–194 kbp capable of encoding 150–170 predicted proteins [27]. They present enveloped icosahedral particles that are approximately 200 nm in diameter (Table 1) [28]. Similar to poxviruses, their replication cycle occurs in the host cell’s cytoplasm and presents some independence in the transcription process. It is also capable of encoding RNA polymerase, poly(A) polymerase, and mRNA capping enzyme. Its transcriptional machinery resembles that of its host. Therefore, it also possesses RNA polymerase containing eight subunits, RNA ligase, capping enzyme, mRNA decapping enzyme, initial transcription factors, elongation factors, histone-like proteins, and topoisomerase [28].

Similar to viruses belonging to the family *Poxviridae*, its transcription also presents a temporal profile. Genes classified as immediate early and early are expressed before the beginning of DNA replication. Then, intermediate and late genes are expressed. This control of gene expression is regulated by promoter motifs and transcription factors [16,28]. Researchers recently mapped transcription initiation sites; they identified them in “upstream” regions of 151 genes present in the ASFV genome [27].

Transcriptomic analysis of genes to classify them temporally, based on RNA-seq and CAGE-seq, identified 101 genes that show differences in their expression in early and late times of infection. Only two time points were evaluated, at 5 and 16 h post-infection (hpi); therefore, the genes were just classified as early or late, according to the evaluated moment of infection. Based on both techniques, 36 genes were classified as early—related to transcription, evasion of the host’s immune response, and DNA replication—and 55 as late—related to transcription, viral structure, morphogenesis, and DNA replication [27]. New transcriptomic analysis of ASFV, considering the entire time range of infection, would bring valuable information about the temporal classification of all genes, establishing a landscape for the transcriptome of this important swine pathogen.

### 3.3. Iridovirus Gene Transcription

The family *Iridoviridae* is divided into two subfamilies. The subfamily *Alphairidovirinae* comprises the genera *Ranavirus*, *Megalocytivirus*, and *Lymphocystivirus*, which are pathogenic to ectothermic vertebrates. The subfamily *Betairidovirinae* is composed of the genera *Iridovirus*, *Chloriridovirus*, and *Decapodiridovirus*, which are pathogens of invertebrates such as insects and crustaceans [29]. Their genomes consist of linear dsDNA that is approximately 105–212 kbp and encodes 92–211 predicted proteins. In addition, they have non-enveloped, icosahedral particles that are 300 nm in diameter (Table 1) [30,51].

Similar to other members of the phylum *Nucleocytoviricota*, their replication occurs partially in the host cell’s cytoplasm. Even with homologous RNA polymerase subunits in the iridoviruses genome, host cell RNA polymerase is necessary to synthesize early gene transcripts of viruses belonging to the *Ranavirus* and *Iridovirus* genera [16,52]. Thus, early mRNAs are synthesized in the nucleus using the host cell’s RNA polymerase II in the early stages. However, it is believed that the late transcripts are synthesized in the cytoplasm of the host cell by the RNA polymerase encoded by the virus [53]. Its gene expression is temporally regulated and results in the expression of three classes of genes: immediate early, early, and late [30,53]. Analyses performed with members of the genus *Iridovirus*, such as red sea bream iridovirus (RSIV), identified nine immediate early genes, 40 early genes, and 38 late genes [54].

The same temporal profile of gene expression regulation has been described for other members of the family *Iridoviridae* and the genus *Ranavirus*, including frog virus 3 (FV-3). This gene has 33 immediate early genes, 22 early genes, and 36 late genes [53]. Although the number of genes belonging to the different time classes is not the same, the temporal profile of gene expression regulation is similar among the members of the family *Iridoviridae*. Immediate early genes are expressed shortly after infection of the host cell and can produce transcripts even in the presence of protein synthesis inhibitors. Among the products of their transcripts, some factors will activate the transcription of early genes. Early genes encode proteins associated with DNA replication, whereas late genes are expressed after viral DNA replication begins and encode the components of the viral particle [54,55].

### 3.4. Ascovirus Gene Transcription

The family *Ascoviridae* consists of two genera. The *Ascovirus* genus includes the species *Heliothis virescens ascovirus 3a* (HvAV-3a), *Spodoptera frugiperda ascovirus 1a* (SfAV-1a), and *Trichoplusia ni ascovirus 2a* (TnAV-2a). The *Toursvirus* genus comprises *Diadromus pulchellus ascovirus 4a* (DpAV-4a) as its only representative species. Viruses belonging to this genus can infect arthropods, mainly lepidopterans, in their larval stage, causing a fatal disease. They have enveloped particles that are 300–400 nm in length and 100–150 in diameter [31,56]. Its genome is circular dsDNA with approximately 116–185 kbp that encode 117–180 predicted proteins (Table 1) [56].

The gene that encodes the major capsid protein has been used to analyze the relationship among the species of the *Ascoviridae* family and other viral families such as *Iridoviridae*. Ascoviruses are believed to have evolved from iridoviruses, based on phylogenetic analyses using the major capsid protein gene and the fact that they share homologous protein sequences. Thus, transcriptional regulation may have been maintained during the evolutionary process [16,32]. Studies on ascoviruses are scarce, especially regarding the processes that regulate the transcription of their genes. Currently, the knowledge we have about ascovirus transcription comes from studies mainly related to the *Ascovirus* genus [33].

Analyses performed using the SfAV-1a virus as a model established temporal transcription classes for the identified genes. Its replication cycle lasts approximately 48 h; thus, 17 genes were classified as early, being expressed at 6 hpi. They have functions related to nucleotide metabolism, inhibition of apoptosis, interaction with the host, and surprisingly, five of these early genes possess functions related to the viral structure. All the early genes analyzed continue to be expressed even at later time points during the infection. Of all the analyzed genes, 44 had their transcription initiated at 12 hpi, being classified as late. These genes present functions related to RNA and DNA metabolism and viral structure, including the major capsid protein and lipid metabolism. In addition, 11 genes expressed at 24 hpi were classified as very late; they are related to viral structure, lipid metabolism, transcription factors, and some with no known function [33].

### 3.5. Phycodnavirus Gene Transcription

The *Phycodnaviridae* family currently consists of the genera *Chlorovirus, Coccolithovirus, Prasinovirus, Prymnesiovirus, Phaeovirus*, and *Raphidovirus*; the members infect a diverse group of eukaryotic algae. These viruses are the primary pathogens of these algae, found mainly in marine environments, responsible for significant control of the host cell population in the oceans. The genome can be linear or circular dsDNA of approximately 180–560 kbp, depending on the specific group of viruses, capable of encoding over 300 predicted proteins, as well as up to 16 genes that encode tRNAs [57,58]. It presents icosahedral particles that vary between 100 and 220 nm in diameter (Table 1). As described for other viruses belonging to the phylum *Nucleocytoviricota*, phycodnaviruses also have a temporal transcriptional profile.

The most studied phycodnaviruses are those belonging to the *Chlorovirus* genus, commonly named chloroviruses. Their replication cycle lasts approximately 6–8 h, and the transcription of its genes can be divided into early and late stages. The transcription of early genes starts 5–10 min after the onset of infection, and the transcription of the late genes begins 60–90 min after the start of DNA synthesis [34,35,36]. Analyses of the transcriptional profile using *Paramecium bursaria* chlorella virus 1 (PBCV-1) as a model identified that 62% of the 365 predicted proteins were expressed at early times. Of the 227 genes expressed before the beginning of viral DNA synthesis, 127 were classified as early and 100 as early/late, which were also detected after viral DNA synthesis. In addition, 133 transcripts were classified as late due to the fact that they were expressed only after the beginning of viral DNA synthesis [35]. Although the transcription of early genes involves transcription factors incorporated into the mature particle, none of the early coded genes that were analyzed were identified as being expressed late and incorporated into the particle. Therefore, these viruses generally depend on the host algae to provide most transcription functions, at least for immediate early expression [35,36].

Unlike other NCLDV members, phycodnaviruses do not encode their own RNA polymerase: they depend on the host cell’s RNA polymerase for the transcription of their genes. Thus, viral DNA and associated proteins migrate to the nucleus to start transcription [37]. However, it has been reported that Emiliania huxleyi virus 86 (EhV-86) has six RNA polymerase subunits, suggesting that this particular member of the Phycodnaviridae family encodes its transcriptional apparatus and might have had a different evolutionary history compared with other relatives [59]. Transcriptomic data on Emiliania huxleyi virus 201 (EhV-201) showed that all six RNA polymerase subunits are expressed at distinct levels, but all at an early or early-late stage of the virus life cycle, possibly being key regulators of the transcription of late genes [60].

Despite having essential elements for transcription, most *Phycodnaviridae* family members do not have their own transcriptional apparatus. The diversity within *Phycodnaviridae* is vast and, despite being a monophyletic group, it appears that each group within the family had a distinct evolutionary pathway, probably related to the hosts to which they are associated. Such differences may point to a reorganization of the family at the taxonomic level in the future [61]. However, the regulation of transcription in this family as well as other NCLDV presents a temporal profile that has been conserved during the evolutionary process.

### 3.6. Transcription of Mimiviruses and Other Giant Amoeba Viruses

In 2003, the discovery of the first giant amoeba virus, *Acanthamoeba polyphaga* mimivirus (APMV), led to the establishment of the *Mimiviridae* family, also belonging to the NCLDV group [38]. Viruses belonging to this family have characteristics never before described in the virosphere, such as particles ~700 nm in diameter, capable of being visualized by optical microscopy. In addition, they have in their particles a layer of fibrils (~120 nm in length) suggested to be immersed in a matrix of peptidoglycan [62]. They also have a face named star-gate, related to the release of the genome in the host cell’s cytoplasm [39]. Their genome is composed of a dsDNA molecule of approximately 1.2 megabase pairs (Mbp), capable of encoding more than 1000 predicted proteins, and a wide range of elements related to transcription (Table 1) [63].

As with other NCLDV, mimiviruses also have a temporal transcription profile in which the transcribed genes are classified as early, intermediate, and late [40]. Genes classified as early are expressed from 0 to 3 hpi. This class of genes is functionally diverse, but most of them are composed of genes with functions not yet determined. Three aminoacyl-tRNA synthetases—TyrRS (L124), MetRS (R639), and ArgRS (R663)—belong to this temporal class and researchers suggest they are involved in viral protein translation from the beginning of the multiplication cycle. In addition, enzymes involved in DNA repair are also expressed during the early stages of the multiplication cycle [40,64], whereas genes classified as intermediate are expressed between 3 and 6 hpi; most of these genes encode proteins involved in the DNA replication stage, such as DNA polymerase and late gene transcription factors.

Finally, genes classified as late are expressed 6 hpi. These genes encode the structural elements such as the major capsid protein and enzymes present in the particles that are necessary at the beginning of the infection. Some topoisomerases, important in DNA replication and encapsidation, are also part of the late gene class. Products related to the expression of late genes have been detected through proteomics analyses in more significant proportions than those of other temporal classes [40,63,64]. Due to the interest of the scientific community in this group of viruses, many other giant viruses have been isolated over the years, identifying a large part of the pathogens of amoebae that had never been investigated. For example, in 2009 researchers isolated a new giant virus in a culture of *A. polyphaga* named *Marseillevirus marseillevirus* (MRSV) from water samples collected in cooling towers in Paris, France. Due to the fact that they present distinct characteristics of mimiviruses, possessing an icosahedral symmetry capsid approximately 250 nm in diameter and a completely new genome, the new viral family *Marseilleviridae* was created (Table 1) [41].

Similar to mimiviruses, Marseilleviruses also have a temporal transcription profile. Analyses have demonstrated that of the 457 genes identified from Marseilleviruses, 83 (18%) are classified as early (0–1 hpi), 218 (48%) as intermediate (1–2 hpi), and 156 (36%) as late (4 hpi). Of all the analyzed genes, 316 have no known function, 36 are related to DNA replication and repair and 25 are related to signal transduction regulation. In addition, genes related to nucleotide metabolism (8 genes), transcription (13 genes), translation (4 genes), and viral structure and morphogenesis (10 genes) have been identified [42].

After the discovery of the first giant viruses and the increasing interest of the scientific community in this group, a great diversity of other giant viruses have been isolated, including faustovirus, pandoravirus, pithovirus, and mollivirus, among others. These giant viruses possess particles ranging from 200 to 1500 nm in diameter and a genome from 466 kbp to 2.5 Mbp encoding 450–2500 predicted proteins (Table 1) [6,7,10,11]. These viruses have a set of identified transcriptional elements, including subunits of RNA polymerase, which suggests that these viruses have greater autonomy in this process. Interestingly, medusavirus, a recently isolated and characterized amoeba virus, has no homolog of RNA polymerase and seems to have a greater dependence on the host for gene transcription [14]. However, this virus has other components of the transcriptional apparatus, including transcription factors SII and VLTF3, which are expressed during the virus life cycle [65]. Regarding other relatives, the gene expression of medusavirus is also temporally regulated, with most of the genes related to DNA replication being expressed at early and intermediate times, while genes involved in virion morphogenesis are expressed at intermediate and late moments of the virus replication cycle [65]. Studies aiming to elucidate how gene expression is regulated in most amoeba viruses are still needed. Despite having similar temporal regulation profiles that can indicate a common origin, studies providing information about the functions of genes belonging to different temporal classes and whether this profile is also maintained among the different viral families belonging to NCLDV are required.

## 4. Functional Comparative Analysis of Temporally Expressed Genes from Pathogens of *Nucleocytoviricota*

It is clear that different members of *Nucleocytoviricota* have a similar pattern of gene expression. However, some questions deserve closer attention. Are the functions observed in genes expressed in different temporal classes similar among different NCLDV? Which functions are shared and predominant in different temporal classes of genes? To answer those questions and to obtain a general picture of this scenario, we performed a functional comparative analysis of the transcription profile and the genes expressed by different NCLDV using the BLASTp and InterProScan tools. We only included viruses whose transcriptome data related to the full extension of the replication cycle and covering all genes were available to allow a proper comparison of the data: MRSV (GenBank accession number NC_013756.1), APMV (GenBank accession number NC_014649.1), VACV (GenBank accession number AY243312.1), FV3 (GenBank accession number AY548484.1), EhV-201 (GenBank accession number JF974311.1) and medusavirus (GenBank accession number AP018495.1). We obtained all the CDSs whose transcriptomic data were available and performed a re-annotation considering the best hit of BLASTp against non-redundant (nr) protein database with an e-value cutoff of 1 × 10^−5^, and searching for conserved domains by using default parameters of InterProScan. After gene re-annotation, we classified them into functional groups compared with the NCVOG categories [1]. To obtain a clear picture and to allow a proper comparison among the viruses, we classified the genes into the following temporal classes: early, intermediate and late, considering the previous classification [25,40,42,53,56,60]. Here it is important to note that the replication cycle of each virus has a different time range. Considering this particular point, we classified the genes into three distinct temporal classes—early, intermediate, and late—based on the moment when they are expressed according to transcriptome data instead of the exact time (hours) of expression to avoid misinterpretation of the data. It is also important to note that in the original works, the authors may have defined different temporal classes for the virus (e.g., *Emiliania huxleyi* virus and medusavirus), where genes expressed very early in the replication cycle were identified and defined as “immediate-early” genes. Nevertheless, these are genes expressed in the initial (early) phase of the virus replication cycle. Thus, for a proper comparison, we decided to group the genes considered “immediate-early” and “early” into a single category, named “early” genes. A similar rationale was used for the reclassification of genes of other viruses into the three temporal patterns adopted in this study. For viruses that had only three temporal patterns defined, the patterns were maintained considering the three phases of the replicative cycle. This strategy allowed a proper comparison among viruses whose subdivision in genes expressed in different phases of the replication cycle was not originally defined. For a comparison between the original classification and that adopted in this study, see Appendix A.

Marseillevirus T19 has 83 coding sequences (CDS) classified as early genes (18%), 218 (48%) as intermediate, and 156 (34%) as late (Figure 1A). The main functions observed for early genes of Marseillevirus are related to DNA replication and recombination, transcription and signal transduction, and some metabolic functions. Of the 83 genes classified as early, 3 (3.6%) are related to DNA replication and recombination. Most genes related to DNA replication are expressed at intermediate or late times during infection; however, some can be expressed as soon as the particle is internalized by the host cell. In addition, 7 (8.4%) genes are related to the transcription process, such as those encoding helicase and RNA polymerase subunits, 13 (15.6%) with regulation and signal transduction similar to some kinases, and only 1 (1%) is related to nucleotide metabolism (Figure 2, Appendix A). The transcription of early genes related to signal regulation and transduction, which encode serine/threonine kinase, suggests that the virus has the potential to manipulate its host’s responses, facilitating the establishment of productive infections. Regarding genes classified as intermediate, the main functions observed are also related to DNA replication and recombination in (25/218, 11.5%), followed by signal transduction regulation (8/218, 3.7%). Interestingly, many genes involved in DNA metabolism are also expressed during the late stages of the replication cycle, suggesting a continuous process of DNA manipulation during the virus life cycle. All of the genes related to virion structure and morphogenesis are expressed late, an expected feature considering that these genes are involved in the formation of new viral particles (Figure 2, Appendix A).

APMV has a total of 979 CDS, but for this analysis we included only genes for which transcriptome data were available, for a total of 829 genes [37]. The remaining 150 genes remain to be classified as soon as new data are obtained. It is important to note that after APMV genome re-annotation [64], some genes initially predicted as complete coding sequences are now considered a single CDS. In that sense, a few differences in the number of genes originally classified into distinct temporal classes and the one adopted in this study occurred (Appendix A). Approximately a third of APMV genes (292, 35.2%) are expressed early, 210 (25.3%) are intermediate and 327 (39.5%) are expressed in the final moments of the virus life cycle (Figure 1B). Most of the genes related to DNA replication, recombination, and repair are classified as intermediate genes, but there are others expressed at other moments in the replication cycle, including DNA primase and some helicases (Figure 2, Appendix A). Genes involved in transcription and RNA processing, including transcription factors, are expressed early, while RNA polymerase subunits are predominantly expressed at intermediate and late times, possibly being important for further viral transcript synthesis. APMV encodes at least 31 genes involved in signal transduction regulation, including F-box domain-containing proteins and serine/threonine kinases, which are evenly distributed among distinct temporal classes of expression. A distinct feature of the APMV is the presence of translation-related genes, including aminoacyl-tRNA synthetases (aaRS) [36]. A total of eight CDS are members of the translational apparatus of APMV, with four being early expressed, including three aaRS, one intermediate (the remaining aaRS fall into this category, i.e., cysteinyl-tRNA synthetase), and three late genes (Figure 2). Similar to MRSV, most genes involved in the mimivirion structure are classified as late genes (14/18, 77.7%) (Figure 2, Appendix A). It is interesting to note that most of these genes are putative membrane genes, possibly involved in the formation of a virus factory and for establishing the initial steps of virion morphogenesis. The gene coding for the major capsid protein (L425) is a late gene.

Different to MRSV and APMV, VACV infects mammals and has been one of the most studied viruses throughout history. Among the 218 genes, 118 (54.1%) are expressed at the initial moments of the replication cycle, while 51 (23.4%) and 38 (17.4%) are intermediate and late genes, respectively (Figure 1C). A few VACV genes are duplicated in the terminal regions of the genome and we included these genes in the analysis as early genes (Appendix A), considering that the corresponding genes were originally classified in this temporal class [25]. In addition, we did not find two intermediate genes (153.5 and 69.5) on NCBI database, thus we included 51 of the 53 originally classified intermediate genes of VACV. It is interesting to note that VACV has many known genes related to the host–virus interaction, most of which (16/19, 84.2%) are expressed during the early moments of the virus life cycle (Figure 2). Among these genes are those related to the host immune response interaction, such as soluble interferon-alpha/beta receptor and chemokine-binding proteins. Most of the genes involved in the transcription and RNA processing of VACV are expressed early (18/24), while only four and two genes are expressed at intermediate and late phases of the replicative cycle, respectively (Figure 2, Appendix A). Genes involved in DNA replication, recombination, and repair are mostly expressed at the early and intermediate phases, and most genes related to virion morphogenesis are classified as intermediate and late genes (Figure 2, Appendix A). Although the functions of many VACV genes have been predicted, almost 40% of the genome remains uncharacterized and many genes have yet to be functionally evaluated, including several genes that have been annotated as ankyrin repeat domain-containing proteins that are similar to other NCLDV.

Among NCLDV included in this analysis, FV3 has the smallest genome and, consequently, fewer genes. Of the 91 genes analyzed, 33 (36.3%) are classified as early, 22 (24.2%) as intermediate and 36 (39.5%) as late (Figure 1D). Half of the FV3 genes have no known function. Ten genes are involved in DNA replication, recombination, and repair, with five being classified as intermediate, two as early, and three as late (Figure 2, Appendix A). Only four genes are involved with transcription and RNA processing, including two RNA polymerase subunits and transcription elongation factor SII, all classified as intermediate genes, and the VLTF3, expressed in the late phase of the replicative cycle. We did not observe genes involved in translation or the host–virus interaction (Figure 2). It is important to note that the absence of genes involved in direct interaction with the host as well as the fact that only a few are involved in signal transduction regulation (two early genes, four late genes) do not indicate that FV-3 only weakly manipulates the host cell. On the contrary, these data should be interpreted with caution given the large number of genes that have not been characterized, and genes involved in the manipulation of host metabolism could be identified with further studies. Curiously, most of the genes involved in virion structure, annotated as surface proteins, are classified as early genes (9/18, 50%), while only 4 (22.2%) are classified as late (Figure 2, Appendix A). This is in sharp contrast to other NCLDV, where genes involved in the virion structure and morphogenesis are mostly classified as intermediate or late genes. Nevertheless, the gene encoding the conserved major capsid protein is expressed late, as observed for most other NCLDV (Figure 3A).

EhV-201 encodes 447 genes and single-cell RNA-seq data indicate that all genes are expressed throughout the replication cycle evaluated from 0 to 24 hpi [34]. These genes are expressed at different moments, which we defined as early, intermediate and late. However, due to the limited data, 52 genes (11.6%) could not be assigned to a category in the original study. Therefore, for our analysis, we considered only the genes that had been confidently classified into distinct temporal classes. From these 395 genes, 90 (22.8%) are expressed early, 185 (46.8%) are intermediate and 120 (30.4%) are late (Figure 1E). The large majority of EhV-201 genes have no known function (374/447, 83.7%). Among the 73 genes with defined functions, 3 could not be included in any temporal class. Genes related to DNA replication, recombination, and repair are mainly expressed at intermediate moments of the replication cycle, similarly to other NCLDV, while those related to the virion structure and morphogenesis are expressed late (Figure 2, Appendix A). Most genes of the transcriptional apparatus of EhV-201, including all six RNA polymerase subunits, are considered intermediate genes, with only one transcription factor, VLTF2, being expressed early (Figure 2, Appendix A). Curiously, genes whose products are involved in lipid and protein metabolism—for example, lipases and proteases—are most expressed early, suggesting a putative role affecting the host’s metabolism.

Researchers recently evaluated the transcriptional landscape of medusavirus by using RNA-seq; this virus was isolated from hot spring water in Japan and contains 461 protein-coding genes [14]. Of these genes, 131 (28.4%) are considered early, being expressed between 0 and 2 hpi; 272 (59%) are intermediate, expressed between 2 and 4 hpi; and 58 (12.6%) are late, expressed after 4 hpi, with higher expression after 8 hpi [65] (Figure 1F). Similar to other NCLDV, the majority of medusavirus genes are uncharacterized (359/361, 77.9%). Most genes related to DNA metabolism are expressed until 4 hpi, including DNA polymerase and viral homologs for histone proteins (Figure 2, Appendix A). Despite the lack of RNA polymerase, medusavirus contains some genes involved in the transcription process, such as transcription factors that are classified as intermediate genes. Interestingly, the viral poly-A polymerase is early expressed and might be related to the poly-adenylation of viral transcripts during the replication cycle. Genes involved in signal transduction are evenly distributed in the three temporal classes, suggesting a constant interaction with the metabolic pathways of the host (Figure 2). Finally, it is curious that, differently from other NCLDV, most of the genes associated with the medusavirion structure and morphogenesis are classified as intermediate genes, with only a putative membrane protein being expressed late (Figure 2). This profile differs from those observed in other NCLDV and a deeper investigation regarding the morphogenesis of this virus could bring important novelties for the field.

As mentioned before, we reclassified the viral genes into three different temporal patterns of expression, considering the different phases of a viral cycle, that is the beginning (early), middle (intermediate), and ending (late), so we could make a proper comparison among the six distinct viruses. However, more than three expression patterns can exist, as originally evidenced for some viruses [60,65]. These additional patterns are probably related to specific regulatory elements, including promoter motifs and transcription factors. It is worth noting that other viruses outside NCLDVs, including herpesviruses, baculoviruses, and bacteriophages, also have similar patterns of gene expression [66,67,68]. Usually, these expression patterns are defined by clustering methods, considering similar profiles of expression levels of the genes, which can indicate the existence of different temporal divisions, despite being expressed in the same phase of the replicative cycle. It results in the definition of different categories by distinct authors. We do not intend to establish a general concept here, but it is important to mention that for a proper comparison, we must consider the different phases of the replicative cycle instead of the different expression patterns defined by clustering methods.

The six viruses included in this comparative analysis exhibit differences in many aspects, including viral structure, genome, and host range. Nevertheless, many studies evidence their common origin [69,70,71,72]. The most recent hypothesis indicates that *Nucleocytoviricota* members evolved from a small group of viruses (*Tectiviridae*), most likely following an accordion-like model of genomic evolution [71,73]. Despite the differences, a common feature among these viruses is the temporal regulatory pattern for gene expression, as revised in this study. Our analysis indicates that despite differences in the genetic arsenal, genes involved in the same biological process are generally expressed in the same phase of the replicative cycle of a virus. A striking divergence is the genes related to other metabolic functions, which are expressed in different phases depending on the virus. It could be due to the direct relationship with the host, where the virus would manipulate the host metabolism, which is specific for each group of eukaryotic organisms. These viral genes were possibly obtained by horizontal gene transfer (HGT) from the host throughout evolution, thus not following a similar expression pattern. Interestingly, for the most conserved genes (i.e., the six genes observed in 95% of all NCLDVs [4]), the expression pattern is similar for all the six viruses, despite some differences that can be observed mainly for the vaccinia virus’ genes, which are majorly early expressed (Figure 3A). That may remain true for other NCLDVs and even their distant tectiviruses relatives. A transcription cascade model of gene expression has been evidenced for tectiviruses [74], reinforcing the hypothesis that the common ancestor of these viruses and NCLDVs already had that mechanism of gene regulation.

Altogether, our findings highlight a common feature among members of *Nucleocytoviricota*: a temporal pattern of gene expression. In general, most of the genes with predicted functions fall into the early (323 genes) and intermediate (325 genes) class of temporal expression, independently of their functional classification (Figure 3B). An expected exception is the genes related to the virion structure and morphogenesis, most of which are classified as late genes. Genes belonging to different functional classifications are unevenly distributed among the viruses included in this analysis; we can expect that such a result remains true when other NCLDV are considered (Figure 3C, Appendix A). Despite a common pattern for temporal gene expression, a feature possibly inherited from the last common ancestor of *Nucleocytoviricota*, it is most likely that each viral group has had a different history of gene gain and loss [75], possibly due to interaction with distinct hosts and sympatric organisms, resulting in different proportions of functional categories of orthologous genes. It is not completely clear whether the transcriptional regulatory patterns observed for NCLDVs are different from other viruses. Different dsDNA viruses, including both large and small representatives (e.g., herpesvirus and bacteriophages), have been described as having a transcription cascade model of gene expression [66,68,76]. Here we only compared the transcriptional patterns of viruses related to six different taxa, all belonging to the phylum *Nucleocytoviricota.* In this sense, it is not possible to claim any exclusivity for the NCLDVs concerning the patterns of gene regulation. Instead, we might expect that such a model of transcription regulation was broadly selected throughout the evolution for different groups of dsDNA viruses, being a striking example of convergent evolution. A more in-depth comparison of the underlying regulatory mechanisms of gene expression among distinct groups of dsDNA viruses could bring valuable insights into their evolution.

## 5. Conclusions and Perspectives

Many concepts about the virosphere have changed with studies carried out over the years following the discovery of NCLDV. This group shares many genes related to the replication of the genome and the formation of the viral structure, called “viral hallmark genes”, which point to the monophyly of this group. In addition, many members of this group have a nearly complete transcriptional apparatus, which provides some independence from their hosts’ machinery. Thus, the presence of a robust transcriptional apparatus has raised much discussion about the evolutionary aspects of these viruses and their genome.

In this work, we performed a comparative analysis of groups of genes expressed at different times of infection of different members of the NCLDV group. We observed that a common characteristic of this group is a temporal expression profile of their genes throughout the replication cycle, a characteristic that has been maintained throughout the evolution. Overall, genes related to genome transcription and replication are generally expressed in the initial/middle phase of the replicative cycle, while those associated with virion morphogenesis and structure are mainly expressed in the final phase of the virus life cycle. Understanding how the genes of a given pathogen are expressed provides data that assist researchers in understanding their biology and interaction with their hosts. In addition, information regarding the regulation of the expression of these genes can also assist in studies with the objective of interrupting this process at a certain point in the cycle to contribute to the resolution of possible diseases caused by different viral pathogens. Finally, this study compiles information about the regulation of gene expression of different pathogens that opens up the field for transcription studies of other NCLDV, for which this process has not been completely elucidated. The analysis presented here provides insights into the gene expression profiles of other viral pathogens belonging to *Nucleocytoviricota* and can be used as a starting point for future transcriptomic investigations.

## Figures and Tables

**Figure 1 pathogens-10-00935-f001:**
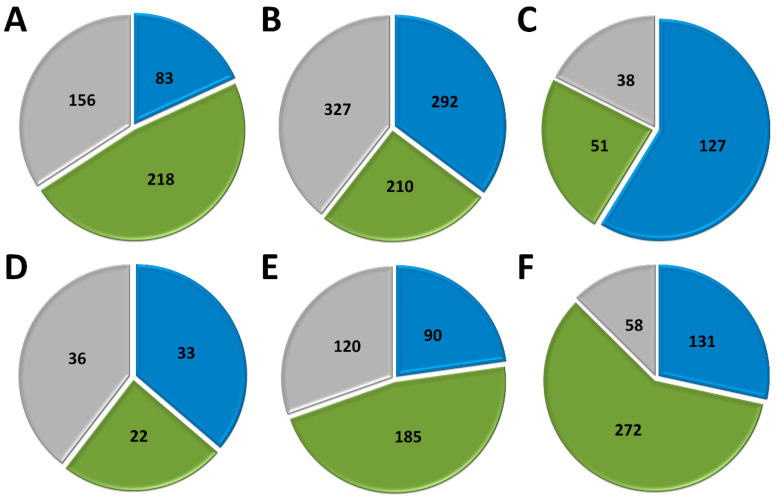
Temporal expression profile of viral genes. Genes for each virus were classified according to the phase of the replicative cycle they are expressed, named early (blue), intermediate (green), and late (grey). The numbers of genes associated to each temporal class are indicated. (**A**) Marseillevirus (*Marseilleviridae*); (**B**) Mimivirus (*Mimiviridae)*; (**C**) Vaccinia virus (*Poxviridae*); (**D**) Frog virus 3 (*Iridoviridae*); (**E**) Emiliania huxleyi virus 201 (*Phycodnaviridae*); (**F**) Medusavirus (Medusaviridae).

**Figure 2 pathogens-10-00935-f002:**
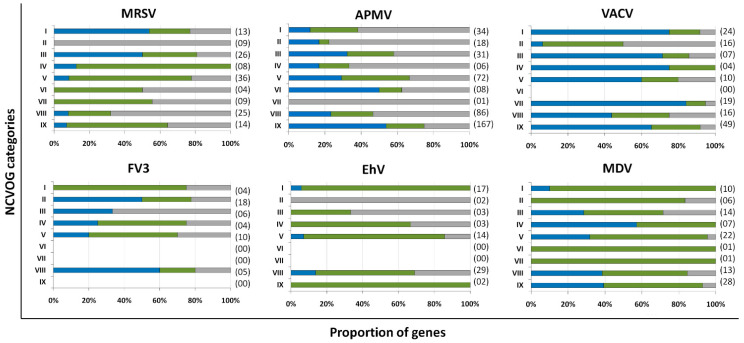
Distribution of genes from different functional categories into temporal gene expression classes. The x-axes contain the different functional categories based on the NCVOG classification. The y-axes represent the proportion of genes for each functional category for all six viruses included in the analysis. Data are shown as a percentage to allow for proper comparison. The number of genes associated with each category is indicated in parenthesis. Blue, early genes; green, intermediate genes; grey, late genes. I, transcription and RNA processing; II, virion structure and morphogenesis; III, signal transduction; IV, nucleotide metabolism; V, DNA replication, recombination and repair; VI, translation; VII, host–virus interaction; VIII, other metabolic functions; IX, miscellaneous. APMV, mimivirus; MRSV, Marseillevirus; VACV, vaccinia virus; FV3, frog virus 3; EhV, Emiliania huxleyi virus 201; MDV, medusavirus. The complete data are included in Appendix A.

**Figure 3 pathogens-10-00935-f003:**
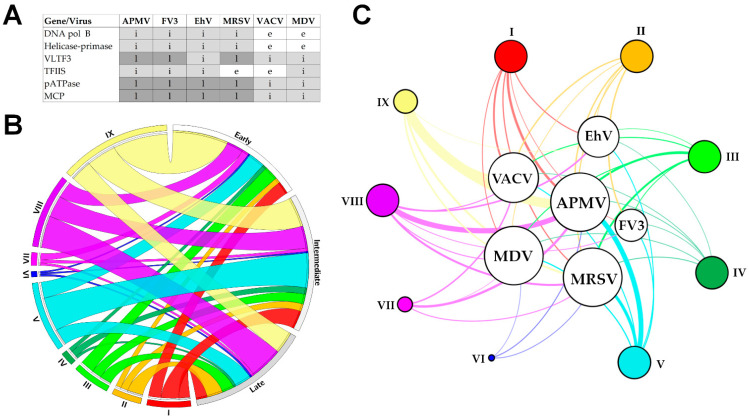
Distribution of functional categories of genes among distinct times of expression and nucleocytoplasmic large DNA viruses (NCLDV). (**A**) Temporal expression pattern of the six most conserved genes among NCLDVs. (**B**) A Circos plot representing the proportion of different functional classes of genes and their respective moment of expression considering the viruses’ life cycle (early, intermediate and late). (**C**) A bipartite network graph connecting different viruses by distinct functional categories of genes. The node diameters are proportional to the edge degree. The edge thicknesses are proportional to the number of genes belonging to each functional class identified in the genome of each virus. The network graph was built using Gephi version 0.9.2. The layout was generated using a force-based algorithm followed by manual rearrangement for better visualization of the connections. Abbreviations: e: early; i: intermediate; l: late; (I–IX) Functional classes of genes: I, transcription and RNA processing; II, virion structure and morphogenesis; III, signal transduction; IV, nucleotide metabolism; V, DNA replication, recombination and repair; VI, translation; VII, host–virus interaction; VIII, other metabolic functions; IX, miscellaneous. APMV, mimivirus; MRSV, Marseillevirus; VACV, vaccinia virus; FV3, frog virus 3; EhV, Emiliania huxleyi virus 201; MDV, medusavirus.

**Table 1 pathogens-10-00935-t001:** General information and transcriptional classification of *Nucleocytoviricota* representative members of different families.

Family	Virus	Particle Size (nm)	Genome Size (Kbp)	Proteins Encoded	Temporal Classification	Time Range	Host Range of the Family	Refs.
*Poxiviridae*	VACV	200–300	195	218	E, I, L	E (0.5–1 h); I (1–2 h); L (4 h)	Mammals, birds, reptiles, fish, insects	[22,23,26]
*Asfarviridae*	ASFV	200	170	152	IE, E, I, L	NA	Pigs and wild boars	[27,28]
*Iridoviridae*	FV-3	300	105	91	E, I, L	E (2 h); I (4 h); L (9 h)	Reptiles, fish, insects, crustaceans	[29,30]
*Ascoviridae*	HVaV-3g	300–400	199	194	E, L, VL	NA	*Arthropods*	[31,32,33]
*Phycodnaviridae*	PBCV	100–220	350	376	E, L	E (5–10 min); L (60–90 min)	Eukaryotic algae	[34,35,36,37]
*Mimiviridae*	APMV	750	1180	979	E, I, L	E (0–3 h) I (3–6 h) L (>6 h)	*Acanthamoeba* sp.	[38,39,40]
*Marseilleviridae*	MRSV	200–250	368	457	E, I, L	E (0–1 h) I (1–2 h) L (>4 h)	*Acanthamoeba* sp.	[41,42]
Pandoraviridae *	PANDV	1000	2470	1430	NA	NA	*Acanthamoeba* sp.	[6]
Pithoviridae *	PITHV	1500	610	467	NA	NA	*Acanthamoeba* sp.	[11]
Molliviridae *	MOLLV	600	651	523	NA	NA	*Acanthamoeba* sp.	[10]

NA: data not available; IE: immediately early; E: early; I: intermediate; L: late; VL: very late. * Taxa proposal. Classification not officially recognized by ICTV. VACV: Vaccinia virus; ASFV: African swine fever virus; FV-3: Frog virus 3; HVaV-3g: Heliothis virescens ascovirus 3g; PBCV: Paramecium bursaria chlorella virus NSy1; APMV: Acanthamoeba polyphaga mimivirus; MRSV: Marseillevirus T19; PAND: Pandoravirus salinus; PITHV: Pithovirus sibericum; MOLLV: Mollivirus sibericum.

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
