# Peer review of "Comparative Analysis of Transcriptional Regulation Patterns: Understanding the Gene Expression Profile in Nucleocytoviricota"

_pathogens, 2021, doi:10.3390/pathogens10080935_

Round 1

Reviewer 1 Report

Review

This article by Gil de Souza et al. provides a review on the studies of gene expression for NCLDVs and then presents a comparative meta-analysis of the transcriptomic landscapes of several NCLDVs. The review part describes the gene expression of Poxviridae, Affarviridae, Iridoviridae, Ascoviridae, Phycodnaviridae and Mimiviridae and other amoeba infecting viruses. The meta-analysis part compares marseillevirus, mimivirus, vaccinia virus, frog virus 3 (an iridovirus), EhV, and medusavirus. Given the accumulation of transcriptomic data for these members of NCLDVs, such an analysis is a timely one. However, unfortunately, the analysis remains largely descriptive and does not well highlight the similarities and differences in the transcriptional cascade, and their origin, between compared viruses.

It is not very clear how the authors used published transcriptome data to classify genes into three categories: early, intermediate and late. For instance, in the original publication, transcriptional profiles of viral genes may be classified into more than three categories (ex. five categories for medusavirus). This should be more clearly explained. It would be useful to provide a table that compares original categories and those adopted by the authors. This comparison should be accompanied with the timing of transcription of each category and the number of genes belonging to each category (in the original and authors classification). Also, is the classification into three categories justifiable? In some viruses there seem to be more than three categories. Why all viral genes shuold be classified into three patterns? Justification or caution in this simplified classification should be carefully argued.

Overall, the transcriptional cascade is one of the main themes of these viral gene expression during the infection. Therefore, how the cascade is realized by the host and viral encoded transcriptional factor should be described with more detail especially for those well studied viruses such as poxviruses.

It is not clear which data support the monophyletic origin of NCLDV from the meta-analysis. I am not sure if such a cascade of gene expression is unique to NCLDVs. The authors should carefully describe the peculiarity of gene expression regulation of NCLDVs especially in comparison with other viruses outside NCLDVs. Otherwise, the clam that the authors obtained an additional evidence for the monophyletic origin is not clear for me and is not supported.

Probably, it is better to put the panels A of Fig. 1 to 6 in one figure so that readers can easily compare them. This also applies to the panels B of Fig. 1 to 6.

Additional minor comments:

L.22 “contributes” -> “contributes to”?

L.32-34: “From this set of genes, .. this group [2,3].” This sentence is unclear.

Official viral taxonomic names such as Nucleocytoviricota, Poxviridae, Asfarviridae, etc. should be italicized.

L.46 “Another characteristic … transcriptional machinery [16].” This sentence is over simplified the situation for several viruses (e.g. Phycodnaviridae and Medusaviridae), which likely depend on the host transcriptional machinery.

L.48 Related to “The presence of robust …[16]”. I suggest to cite Julien Guglielmini et al 2019 (DOI: 10.1073/pnas.1912006116).

L.77 “for different times” -> “for different length of time”?

L.78 “at similar moments” is awkward.

L.95 “tecnologies” -> “technologies”

L.134 “leaving the intermediate gene sequence available for viral RNA polymerase”. I am not sure if I understand this. Maybe this should read “leaving the early genes available for viral RNA polymerase”?

L.212 “encode the viral particle” -> “encode the components of viral particles”?

L.237 “at late times of infection” -> “at later time points during infection”?

L.273-276: Which viruses are you referring to here is unclear, especially you describe about EhV in the previous section, while the you cite Chlorovirus studies here.

L.287: “immersed in a matrix of peptidoglycan” could you give a citation to this statement?

L.353: “then” -> “them”

Reviewer 2 Report

This manuscript by Gil de Souza et al. deals with transcriptional patterns and gene expression dynamics during infection by Nucleocytoviricota.

For the most part, the manuscript is well written and straightforward. I do have a few comments:

  1. I think the manuscript will benefit from rearranging so that the common features, as described in figure 7, come before the long list of expression patterns defined in figures 1-6.
  2. As a review, I think a more profound discussion on the biological and evolutionary insights that can be gained from the analyses presented in the manuscript will be beneficial.
  3. The authors describe 40 core genes that are common to all NCLDV. Is the expression pattern of these genes also common? Can this shed light on their evolutionary origin and biological function?
  4. I think the term “Meta analysis” used in the title is a bit strong for comparing transcription profiles of 6 viruses.
  5. There are a few instances of inaccuracy presented in the manuscript. For example, in the first paragraph about the Phycodnavirus, it is stated that their genomes are linear. However, in Wilson et al. (Science, 2005), they describe the genome of EhV as circular. Additionally, the authors claim that phycodnaviruses do not encode their own RNA polymerase (in the third paragraph). However, on page 12, they state that EhV201 encodes for 6 RNA polymerase subunits. These inconsistencies might be because each virus family has many groups that might not all have the same genome structure and gene content. If this is the case, it should be discussed in the review. If there are only outliers, this can also be an interesting point to address and might shed light on the evolution of these viruses.
  6. There are no details about the parameters, scores, or e-values that were applied for the two tools used in the analysis (Blastp and InterProScan).
  7. Please add titles to the X-axes of all figures 1-6.
  8. In the supp. figure, please write the gene categories on each graph.

Round 2

Reviewer 2 Report

The resubmitted manuscript by Gil de Souza et al., is greatly improved. I especially appreciate the rearranged version and the new figures, it greatly improves the reading experience. The only (minor) comment I have refers to the language. There are a few sentences that can be improved with a little editing. I have listed a few (but not all) below:

L34 – Change “viruses belonging to this group known to date” to “all known viruses that belong to this group”.

L147 – “that is synthesized” or “to be synthesized”.

L153-155 – Please rephrase. What does “regulate” refer to? The presence or the motifs?

L399 – the name of the organism should be in italic

L407 – remove “us” or change to “for”.

L571 – “have been evidencing”? not clear.

Author Response

Dear reviewer, thank you very much for all your considerations that helped to improve our manuscript. We are glad that you appreciated our modifications. We now corrected the sentences you pointed out in this revised version of the manuscript. Please, find below a point-by-point response to all your comments:

L34 – Change “viruses belonging to this group known to date” to “all known viruses that belong to this group”.

Answer: Done.

L147 – “that is synthesized” or “to be synthesized”.

Answer: Done.

L153-155 – Please rephrase. What does “regulate” refer to? The presence or the motifs?

Answer: We rephrased the sentence to avoid misunderstandings.

“Intermediate promoter motifs regulate the expression of late transcription factor genes, consistent with the cascade model of regulation.

L399 – the name of the organism should be in italic

Answer: Done.

L407 – remove “us” or change to “for”.

Answer: We removed the word “us”.

L571 – “have been evidencing”? not clear.

Answer: We rephrased the sentence to be more straightforward.

“Nevertheless, many studies evidence their common origin [69-72]”

This manuscript is a resubmission of an earlier submission. The following is a list of the peer review reports and author responses from that submission.

Round 1

Reviewer 1 Report

Summary: The first part of this review describes transcriptional regulation for each family in the phylum Nucleocytoviricota, focusing on what appears as a common theme: the temporal regulation of gene transcription corresponding to stages of viral infection, leading to the delineation of early, intermediate and late gene temporal classes. The second part of the review comprises a meta-analysis in which the predicted functions of genes in each temporal class in a representative marseillevirus, mimivirus, poxvirus and iridovirus are compared.

Broad Comments: The overall question of whether there is a common pattern in transcriptional regulation in the Nucleocytoviricota merits attention and could well be treated by a literature review and the “functional meta-analysis” approach that the authors have adopted. This review provides a fair introduction to the topic in terms of attempting to cover the literature for the entire phylum. However, it lacks a solid conceptual framework and sufficient synthesis of the research field, which makes it difficult to glean much information or insight into the topic. Crucially, while the review repeats that there is a striking commonality between members of Nucleocytoviricota, ie. their genes are transcribed in a “temporal profile”, this is stated as a bald fact with no clear introduction to what temporal transcriptional regulation is or how it is observed and defined. The description of what methods have been used to show temporal transcriptional patterns is superficial and therefore there is little evaluation of different ways temporal gene classes may be defined. For example, the review touches on how early genes were initially defined as those genes expressed before viral genome replication, but how are early genes defined when the timing of genome replication is unknown? How do Microarrays compare to CAGE-Seq or RNA-Seq? How have genes been temporally classified when they show profiles of all viral genes increasing in their level of transcription over time? The lack of this conceptual grounding means that section 2 reads as a series of facts. Similarly, as it is unclear if the studies used in the metaanalysis are comparable in how temporal classes are defined. The authors clearly made a big effort in attempting to format the transcriptomic data of four disparate studies to try and make them comparable. Nonetheless, the presentation of the section 3 meta-analysis is, to me, not readily interpretable and the conclusion that this shows a common evolutionary thread is not credible. Lastly, the treatment of the literature is not sufficiently comprehensive with several instances where there are references that could better represent the state-of-the-art. One example that stands out is that the authors state they only include marseillevirus, mimivirus, poxvirus and iridovirus in their meta-analysis because only these viruses have been studied over the entire infection cycle; yet within their own reference list, there are such studies of an ascovirus and a phycodnavirus — so why were they omitted? The structure of the article needs reworking, particularly the ordering of the information is not coherent (why start with a description of poxviruses?) and as a result, the subsections are unnecessarily repetitive (eg. repeating for every viral family that gene transcription is temporally regulated). The writing also needs to be extensively revised for clarity as there are many sentences where the meaning is not clear.

Specific comments:

* The Introduction is rather disjointed and does not to my view introduce the idea of transcriptional regulation. It begins with 2 paragraphs about the importance of poxviruses as emerging source of disease, which although interesting does not introduce the core topic of the review. The paragraph introducing transcription in NCLDV is very vague and does not specify the process of differential transcription during infection well or define what temporal regulation is.
L28 “The Poxiviridae family consists of

L29 “which includes species that are pathogenic for humans”

L37 missing comma “hosts, such as”

L38 The sentence it a bit awkward. I suggest “that act as wild animal reservoirs, which sporadically cause infections in humans”.

L39 Also a bit ambiguous. “Thus, these reservoirs impede..”

L45 “…CPXV infections have increased in Africa”

L46 The tense is strange. I suggest “…these populations have become increasingly vulnerable, contributing to…”

L50 Remove “Besides”

L59 “Analyzes performed” is very vague. I suggest “Comparative genomic analyses derived a set of 40 core genes common to the NCLDVs”

L62 “Identified” in the common ancestor implies a direct observation. I suggest “reconstructed as present in the common ancestor seem”.

L67 I suggest “exhibit” instead of “possess”.

L70-71. This sentence is unclear. I suggest “Also, their gene transcription is temporally regulated allowing sets of genes to be classified as early, intermediate and late in accordance with the stage of infection when they are transcribed”.

L71-73. Poorly explained. I suggest “A previous study has divided the genes of the NCLDVs into clusters of orthologous groups (NCVOGs), many of which could be assigned to putative functional classes.”

L75-76 This sentence is ambiguous. What is a comparative functional analysis? What does “these groups of genes” refer to? The genes involved in transcription or the NCVOGs?

L85-90 This information is redundant with the first two paragraphs of the introduction.

L91-96 These sentences are general information better placed in a general introduction. “Viruses possess…transcriptional machinery”.

L96 “The complexity of the poxvirus genome…” This sentence is missing words and I wonder if this is historically accurate. I find it hard to believe genome complexity was what first led researchers to think that poxvirus replication was independent of the host nucleus. Would it not have been more likely the fact one can observe poxvirus assembly in the cytoplasm with a microscope?

L130-132 These sentences are superfluous.

L149 “The start and end of transcription occur at precise positions throughout the genome…” I do not understand this sentence. Does it mean that genes that are co-regulated are found on the same genomic region?

L154 What analysis? How was the classification performed?

L160 This line is a very poor segue to the next section.

L191 Again, this is not a way to segue from one section another. It reads as a very forced statement about evolution that is not well supported.

L326 “BLAST and Interproscan tools” is much too vague as a description of the methods. There needs to be more information about how the genes are functionally classed.

L326-327 “We included those viruses whose transcriptome...full extension of the replication cycle” This does not seem correct as I believe full infection studies are available for other virues (Coccolithovirus, Prasinovirus, Chlorovirus, Ascovirus)

L328-329 The accession numbers given are for genomes, not transcriptomes except for the iridovirus representative (Frog virus 3), which is for a microarray set. However, Frog virus 3 does have a genome (NC_005946.1), Which data were used? Why is it called iridovirus and not Frog virus 3?

L330-332: “To obtain a clear picture…we classified the genes among the following temporal classes: early, intermediate and late, based on previous classification [42,57,60,61]”

The definition of the temporal classes is the foundation of the analysis. It must be described clearly how it was decided to include a set of genes into the early, intermediate or late class. Just saying “previous classification” is not sufficient. Does it mean that the early class for each virus was functionally defined based on the transcriptome profiles?

Table 1

The Table legend does not provide enough information to accurately represent the information in the table. What does “transcriptional classification” mean? Does it mean that gene transcription has only been observed to be regulated in these temporal stages of the infection cycle?

The general information given is not consistent. Sometimes it appears to refer the specific virus in column 2, sometimes it seems to refer members within the family eg. the genome size range for Phycodnaviridae seems to be for the family and not for the species PBCV1. There should be an explanation of whether this information refers to species or family and to remain consistent. Also, references to sources of the general information are not provided.

Replace “Família” with “Family” in header in column 1.

Replace accented “í” in “Virus” in the header in column 2.

The spaces between the commas in the reference lists are not consistent.

Figures 1-4 need to synthesize the results so we can compare between species.

Figure 5, I do not see a similar pattern between the species.